# Regeneration of the larval sea star nervous system by wounding induced respecification to the Sox2 lineage

**Minyan Zheng†, Olga Zueva, Veronica F Hinman***

Department of Biological Sciences, Carnegie Mellon University, Pittsburgh, United States

**Abstract** The ability to restore lost body parts following traumatic injury is a fascinating area of biology that challenges current understanding of the ontogeny of differentiation. The origin of new cells needed to regenerate lost tissue, and whether they are pluripotent or have de- or transdifferentiated, remains one of the most important open questions . Additionally, it is not known whether developmental gene regulatory networks are reused or whether regeneration specific networks are deployed. Echinoderms, including sea stars, have extensive ability for regeneration, however, the technologies for obtaining transgenic echinoderms are limited and tracking cells involved in regeneration, and thus identifying the cellular sources and potencies has proven challenging. In this study, we develop new transgenic tools to follow the fate of populations of cells in the regenerating larva of the sea star *Patiria miniata*. We show that the larval serotonergic nervous system can regenerate following decapitation. Using a BAC-transgenesis approach we show that expression of the pan ectodermal marker, *sox2*, is induced in previously *sox2* minus cells , even when cell division is inhibited. *sox2*+ cells give rise to new *sox4*+ neural precursors that then proceed along an embryonic neurogenesis pathway to reform the anterior nervous systems. sox2+ cells contribute to only neural and ectoderm lineages, indicating that these progenitors maintain their normal, embryonic lineage restriction. This indicates that sea star larval regeneration uses a combination of existing lineage restricted stem cells, as well as respecification of cells into neural lineages, and at least partial reuse of developmental GRNs to regenerate their nervous system.

**\*For correspondence:**
vhinman@andrew.cmu.edu

**Present address:** †Department of Genetics, Harvard Medical School, Boston, United States

**Competing interest:** The authors declare that no competing interests exist.

## Editor's evaluation

This manuscript presents a careful study of nervous system regeneration in the larval sea star using new transgenic tools for marking and following cells involved in regeneration. The authors find that these animals can regenerate their nervous system by the re-specification of existing cells, which are induced to express the embryonic neurogenesis program. The experimental approach is robust and creative and the data interpretation sound. For its contribution to our understanding of how cells are induced to contribute to specific cell lineages during regeneration, this work will be of interest to the broad community of researchers in regenerative and developmental biology.

## Introduction

Regeneration is a fascinating phenomena that challenges the tenet of an irreversible, directional ontogeny. Decades of research have provided extensive understanding of how embryonic cells acquire their fate through the action of GRNs and the sequential turning on, and off, of regulatory genes. During embryogenesis, these developmental GRNs start from a very particular cell type, the single-celled zygote, while in regeneration another source of multi-, or toti-potent cells must be the source of

cells for regeneration. The source of the cells, their molecular state and how, or even whether, developmental GRNs are re-established in these cells are open questions. This knowledge, however, is needed as a prerequisite for understanding the molecular mechanisms of regeneration. Furthermore, an understanding of whether any processes are homologous or if there are taxon-specific mechanisms that are related to the ability for extensive or limited regeneration awaits sufficient sampling across the metazoa.

The deuterostomes, that is vertebrates, invertebrate chordates, echinoderms and hemichordates, are an especially important system to study from the standpoint of regeneration. The vertebrates, with some few exceptions, have limited capacity for regeneration that is usually restricted to specific cells, tissues and organs (*Rodriguez and Kang, 2020*; *Tanaka and Reddien, 2011*). The emerging evidence points to the use of tissue-specific stem cells and corresponding specific niches as a cellular source for regeneration (*Kragl et al., 2009*). Echinoderms on the other hand have an impressive ability for extensive whole body regeneration found in most species of the phylum (*Ferrario et al., 2020*). For this reason, there has been extensive research into the mechanisms of regeneration in many species of echinoderms (*Ferrario et al., 2020*; *Kondo and Akasaka, 2010*; *Dupont and Thorndyke, 2007*; *Piovani et al., 2021*; *Vickery et al., 2001*; *García-Arrarás et al., 2018*; *Mashanov and Zueva, 2019*; *Mashanov et al., 2020*). For example, molecular and ultrastructural studies in brittle star arm regeneration point to use of adjacent epithelial cells to differentiate into sclerocytes needed for arm skeleton regeneration, recapitulating, at least in part, a development program (*Piovani et al., 2021*). Comprehensive studies in the regeneration of radial nerve chords in sea cucumbers also point to the use of radial glial dedifferentiation in nervous system regeneration (*Mashanov and Zueva, 2019*; *Mashanov et al., 2015a*). The emerging consensus from these and other studies is that echinoderms rely extensively on wound-adjacent dedifferentiation for regenerating lost tissues (*Ferrario et al., 2020*; *Mashanov and Zueva, 2019*). Conversely, expression of stem-cell marker genes have been identified as possible cell sources in some species (*Mashanov et al., 2015b*; *Reinardy et al., 2015*). Whether echinoderms and vertebrates use similar cellular sources and mechanisms for regeneration (*Zhang et al., 2003*; *Joven and Simon, 2018*) is important for untangling whether these mechanisms are causative of whole-body versus limited regeneration potential. Understanding these processes in naturally regenerating contexts also has important implications for improved translational applications of induced pluripotent stem cells and other stem cell therapies.

To date, however, while there has been a comprehensive series of studies of the molecular and ultrastructural basis of regeneration in many species of echinoderms, there have been no cell lineage tracking studies. Only cell tracking can definitively establish the origin and trajectory of cells during regeneration and resolve the debate as to the role of stem cells versus cellular reprogramming in echinoderms. Cell tracking, however, is technically challenging, especially in adult echinoderms where there are many cells and cell types, and regeneration proceeds over days or weeks. Transgenic tools are also limited in adult echinoderms. Indeed, the challenges associated with accurately tracking cell lineages have greatly limited the taxonomic distribution of animals for which we have this knowledge across the metazoa.

Here, we sought to fill this important gap by turning to the echinoderm larval regeneration model. Echinoderms, typically for many marine invertebrates, have a biphasic lifestyle, producing a planktonic larval form that exists for up to many months before undergoing dramatic metamorphosis. It has now been well documented that the larval forms are also capable of extensive regeneration (*Oulhen et al., 2016*; *Vickery et al., 2001*; *Kasahara et al., 2019*). We have recently introduced the larval sea star *Patiria miniata* as a model system for the study of whole body regeneration (*Cary et al., 2019b*). Sea stars, as typical for echinoderms, have extensive regenerative capacities, a feature which extends across both their adult and larval life phases (*Vickery et al., 2001*; *Vickery et al., 2002*; *Ben Khadra et al., 2017*; *Hernroth et al., 2010*; *Oulhen et al., 2016*). Although pentaradial as adults, there are many parallels between echinoderm larval development and early vertebrate development, particularly in the formation of the anterior-posterior axis and development of the nervous system (*Angerer et al., 2011*; *Hinman and Burke, 2018*; *Range et al., 2013*). Therefore, while vertebrates and sea stars have very different regenerative capacities, there are commonalities in their early development. There are also extensive genomic resources (*Cary and Hinman, 2017*; *Cary et al., 2019a*, *Cary et al., 2018*; *Cary et al., 2017*; *Gildor et al., 2019*) and detailed developmental GRNs available for *P.*

*miniata* (*Cary et al., 2020*) allows comparisons between developmental and regenerative GRNs and clear gene orthology mapping.

Previous work has shown that when these sea star larvae are decapitated, they are able to regenerate their anterior structures (*Cary et al., 2019b*; *Oulhen et al., 2016*; *Vickery et al., 2001*). Larvae have an anterior serotonergic nervous system, which we show here is removed by this decapitation. Extensive previous work in *P. miniata* embryos has shown how these neurons form during embryogenesis. The SRY-box transcription factor (TF), *sox2* (formerly *soxb1*), is expressed broadly throughout the ectoderm by blastula stage (*Yankura et al., 2010*). *Sox4+* (formerly *soxc*) neural progenitors arise in the ectoderm, and then divide asymmetrically to produce LIM homeodomain TF *lhx2+* (formerly *lhx2/9*) daughter cells, which in turn divide asymmetrically to form differentiated serotonergic neurons (*Cheatle Jarvela et al., 2016*). This occurs within the anterior ectoderm, which expresses TFs such as *six3* and f*oxq2*, and is a *wnt* ligand negative territory (*Cheatle Jarvela et al., 2016*; *Yankura et al., 2013*).

In this study, we took advantage of cell-type-specific markers of differentiated neurons to confirm that the sea star larvae are able to regenerate their anterior nervous system. We establish a novel photoconvertible BAC reporter system to trace populations of cells to determine the cellular origin of these regenerated neurons. This allows us to specifically test whether regenerated neurons arise through embryonic neurogenesis pathways, and to determine the origin and potency of progenitors. Using a small molecule inhibitor of nuclear DNA replication (to question whether new gene expression can arise in the absence of divisions that are a prerequisite of stem cell origin), we show that expression of the putative pluripotency factor *sox2* is induced in previously *sox2* minus cells at the wound site. As this occurs even in the absence of new cell division, this result suggests that these cells are respecified rather than arising from asymmetric stem cell division. Both the newly induced and existing, *sox2+* cells now give rise to new *sox4+* neural precursors that then proceed along an embryonic neurogenesis pathway to reform the anterior nervous systems. The *sox2+* cells contribute to only neural and ectoderm lineages, indicating that these progenitors maintain their normal, embryonic lineage restriction.

## Results

### *Patiria miniata* larvae fully regenerate their nervous system

Sea star bipinnaria larvae have an extensive nervous system (*Katow et al., 2009*; *Murabe et al., 2008*; *Nakajima et al., 2004*; *Carter et al., 2021*; *Elia et al., 2009*; *Hinman and Burke, 2018*). *Figure 1A–B"* shows the 7-day post fertilization (7dpf) *P. miniata* larval nervous system labeled via localization of anti-synaptotagmin B (SynB) and serotonin antibodies. It has been shown that the pan-neuronal marker SynB labels all neurons in many echinoderm species (*Burke et al., 2006*; *Nakajima et al., 2004*). The neural bodies and axonal tracts are apparent throughout the two ciliary bands, the lip of the mouth, domains lateral to the mouth that mark the location of the apical ganglia in this stage, and the esophagus (*Figure 1A*). Serotonin immunoreactivity is concentrated predominantly in the anterior part of the larvae; in the dorsal ganglia with a subset of cells across the aboral surface (*Figure 1B, D' and E'*). The presence of serotonin, characteristic neural cell morphology including long axonal processes and lack of markers of cell division, are taken as evidence that these are differentiated cells.

We have shown previously that larvae are able to regenerate their anterior body when they are bisected below the mouth (*Cary et al., 2019b*; *Figure 1D–I*, and *Figure 1—figure supplement 1*). This bisection, therefore, removes much of the anterior nervous system, including the anterior ciliary band loop (preoral ciliary band), the dorsal ganglia, as well as the mouth and its associated neurons. We tested whether these bisected larvae regenerate their nervous system. By one week post bisection (*Figure 1C–C'*), we find neural bodies and axonal tracts along the newly formed anterior ciliary band and complex neural networks lateral to the mouth suggestive of the dorsal ganglia. The reformed mouth also has associated neurons. We also specifically examined the serotonergic neurons (*Figure 1D'–i*). This neural subtype is found in the dorsal ganglia where they are present as clusters of large cell bodies with long axonal processes (*Figure 1D'–e*). These cells originally form in the anterior-most ectoderm of the late gastrula but migrate posteriorly to a location dorsal to the mouth in the larva (*Cheatle Jarvela et al., 2016*; *Yankura et al., 2013*). Serotonergic neurons are also found on the lower lip of the mouth where they have smaller cell bodies and shorter axonal processes. We therefore

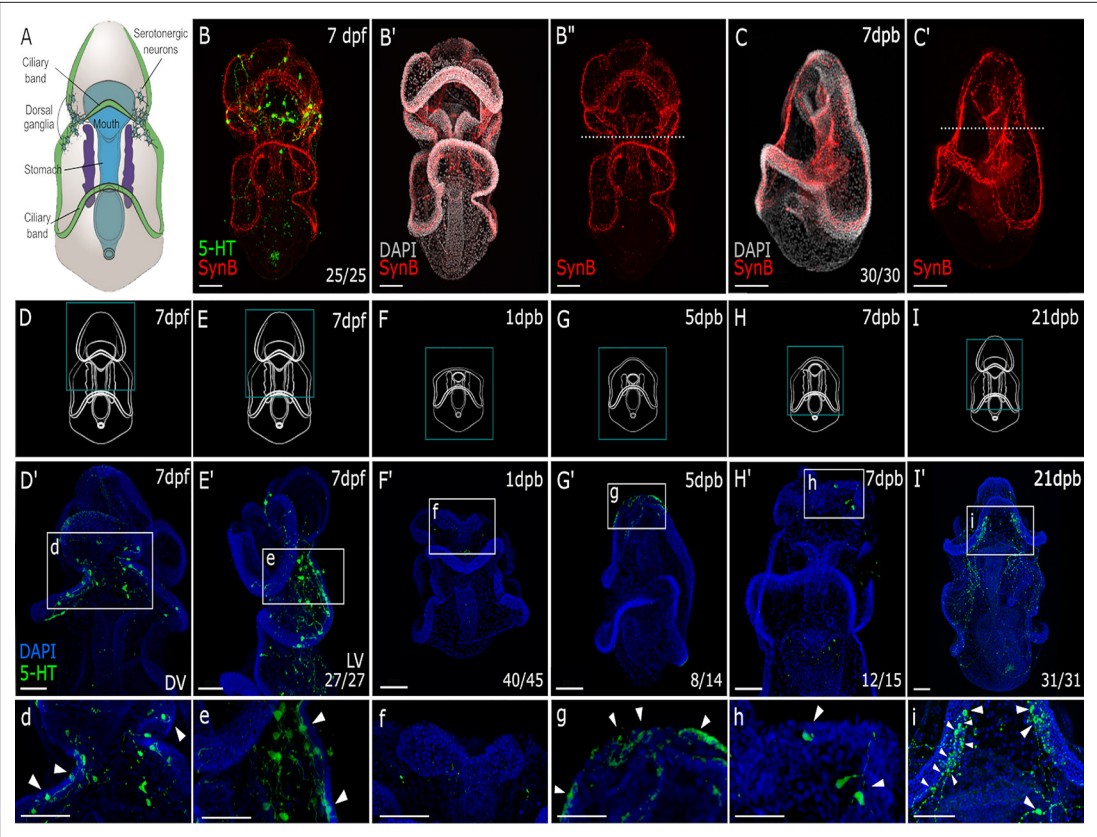

**Figure 1.** Regeneration of the sea star larval nervous system. (**A**) A schematic of a 7-day old *Patiria miniata* larva. (**B-B″**) Immunostaining with anti-synaptotagmin B (SynB; red) and anti-serotonin (5-HT; green) shows the larval nervous system organisation. Serotonergic neurons are distributed along the pre-oral ciliary band, post-oral ciliary bands and around the mouth. Larvae are bisected beneath the lip of the mouth shown by the dotted line (**B″ and C′**). (**C-C″**) A latero-ventral view of regenerated larva at 7 days post bisection, the nervous system stained with Syn B. As bisection was performed at the site represented with the white dashed line. Above the dashed line in (**C′**) is the regenerated anterior tissue with reformed pre-oral ciliary band and the mouth. (**D–I**) Schematics of larvae at the normal (uncut) and regenerated stages corresponding to immunostaining with 5-HT images in (**D′-I′**) and (**d-i**). (**d-i**). The expanded view of the region highlighted with the dashed-line box in (**D′-I′**). (**D′**) The serotonergic neurons are located bilaterally in uncut larvae, on the dorsal side as shown in lateral view in (**E′**). (**d-e**). Neural bodies are embedded in the ectoderm, and project long axonal processes typical of this neural type. (**F′-f**) Bisection removes the serotonergic neurons. (**G′**) By 5 dpb, serotonergic neurons are detected at the regeneration leading edge with emerging neural morphology as shown in (**g**). (**H′**) By 7 dpb, regenerated serotonergic neurons with mature neural morphology (**h**) are located at the lateral side of the regenerated anterior. (**I′-i**). Regenerated serotonergic neurons are bilaterally located to reform the dorsal ganglia by 21 dpb. Arrowheads highlight the serotonergic neurons. Dpf: day-post fertilization; dpb: day-post bisection. DV: dorsal view. LV: lateral view. Scale bar: 50 μm. The numbers shown in the lower right corner of each image indicate the number of larvae showing a positive IHC signal among the larvae examined.

The online version of this article includes the following figure supplement(s) for figure 1:

**Figure supplement 1.** Bright-field micrograph showing sea star larvae undergo whole-body regeneration (WBR).

**Figure supplement 2.** Quantification of serotonergic neurons in larvae shows a stable number of serotonergic neurons over time.

reasoned that serotonergic neural bodies were removed entirely by our manual bisection protocol. Indeed, when we bisect the larvae across the midline, 40 of 45 stained larvae show no remaining serotonergic neurons 1 day later, when they were first assayed. *Figure 1F′–f* shows one such example. Of the remaining five, only one or two cells were found which, due to their location, are likely those normally associated with the lower lip of the mouth.

By examining a time series of regeneration, we show that serotonergic neurons are first revealed by antibody staining at the anterior regenerating leading edge in 8/14 larvae 5 days following bisection (*Figure 1G′–g*). By 7 days post bisection (dpb), serotonergic neurons are found in 12/15 regenerating larvae at the anterior edge (*Figure 1H′–h*). Three weeks following bisection, the regenerated serotonergic neurons are bilaterally patterned at the dorsal side of the regenerating larvae, resembling the dorsal ganglia structure found in intact larvae (*Figure 1I′–i*).

These data therefore show that the anterior serotonergic neurons are removed following anterior bisection and are reformed in the correct general location and with extensive axonal projections by 21 days of regeneration.

## Embryonic neurogenesis pathways re-emerge during regeneration

We next questioned whether the regenerated serotonergic neurons formed using the embryonic neurogenesis pathway. We have shown that, during normal embryogenesis, the SRY-box TF, *sox2*, (formerly *soxb1*) is expressed broadly throughout the neurogenic ectoderm (*Yankura et al., 2013*). Serotonergic neurons form from cells expressing the SRY-box TF, *sox4* (formerly *soxc*) that are first present and distributed in a 'salt and pepper' pattern throughout the ectoderm of the embryonic blastula (*Yankura et al., 2013*; *Yankura et al., 2010*). Two transcription factors, *foxq2* and *six3*, are expressed in the anterior ectoderm of the early gastrula and are required for the correct progression of the anteriorly most located *sox4+* neural progenitors to become *lhx2+* cells. After asymmetric cell divisions, *lhx2+* progenitors in turn give rise to post-mitotic neurons expressing *elav* (*Figure 2A–C*). These post-mitotic neurons will produce serotonin when they mature (*Cheatle Jarvela et al., 2016*). Thus we sought to determine whether these genes are re-expressed following bisection. If so, this can provide the first indication that embryonic neurogenesis is reactivated during regeneration.

Following bisection, we find that *foxq2* and *six3* are expressed in the anterior ectoderm of the leading regenerating edge by 3dpb (*Figure 2D*, and *Figure 2—figure supplement 1A-H*). It is important to note therefore, that *foxq2* is now expressed in cells that, in the normal larvae, reside along the middle of the AP axis and would never normally express anterior markers such as *foxq2*. We find similar patterns with other regulatory genes that are expressed along the embryonic AP axis, with *wnt3* re-expressed in posterior regenerating larvae (*Figure 2—figure supplement 1I-K*). This indicates that the anterior leading edge cells are being respecified, as defined by expressing new sets of genes, during regeneration. This respecification, at least partially, recapitulates the embryonic axial state required for neurogenesis.

*Sox2* is expressed at low levels throughout the ectoderm of the normal larvae (*Yankura et al., 2013*; *Figure 2—figure supplement 2A*) but within 1 day following bisection, *sox2* expression is dramatically up-regulated at the wound site (*Figure 2E–e*, and *Figure 2—figure supplement 2B*), when compared to larval expression levels. We have previously shown that cell division occurs throughout the normal larva, but then decreases in the early bisected larva and is then predominantly localized to the wound adjacent anterior at 3dpb (*Cary et al., 2019b*). Thus, localized *sox2* expression precedes the formation of the proliferative zone but then remains within this anterior region. This regenerative proliferative zone is readily visualized by staining for 5-Ethynyl-2′-deoxyuridine (EdU), by incubating larvae for one to six hours in EdU sea water. This costaining shows that *sox2* is highly expressed throughout this ectodermal zone and within some of these proliferating cells at 3dpb (*Figure 2F–f*, and *Figure 2—figure supplement 2C*) and that this remains through to 5dpb (*Figure 2G–g*, and *Figure 2—figure supplement 2D*).

*Sox4* expression, a marker of neural precursor cells, is also observed at the wound site within one day following bisection (*Figure 2H–h′*, and *Figure 2—figure supplement 2F*). In 3dpb larvae, *sox4* expression is detected at the regeneration leading edge (*Figure 2—figure supplement 2G*). Fluorescent labels allow for the detection of coexpression of genes and colocalization of gene expression with EdU staining. The large number of cells at these stages make colocalization difficult to establish through conventional confocal microscopy and thus we used Imaris Software (Oxford Instruments) 3D visualization analysis to identify overlapping signals with confidence (Imaris visualization is shown *Figure 2h′ and i′* j′, k1, k2). Thus we show that *sox4+* cells clearly undergo cell divisions as FISH staining colocalizes with EdU (*Figure 2I–i′*). Proliferating *sox4+* cells are also detected in the regenerating ectodermal/endodermal mouth (*Figure 2I*). By 5dpb, *sox4* expression is concentrated to the regenerating anterior, again in dividing cells (*Figure 2J–j′*, and *Figure 2—figure supplement 2H*).

The expression of *lhx2* is also detected by 5-7dpb in dispersed cells at the regenerating leading edge (*Figure 2* K-K2, and *Figure 2—figure supplement 2J*). Similarly, the expression of post-mitotic neural marker *elav* is also restored at the regenerating anterior at 5-7dpb (*Figure 2L–l′*, and *Figure 2—figure supplement 2L*). Two-color FISH shows that, similar to embryogenesis, *sox4/lhx2* co-expression is found in cells located at the lateral sides of the regeneration leading edge (*Figure 2K–l′*) and that *lhx2* and *elav* are also co-expressed.

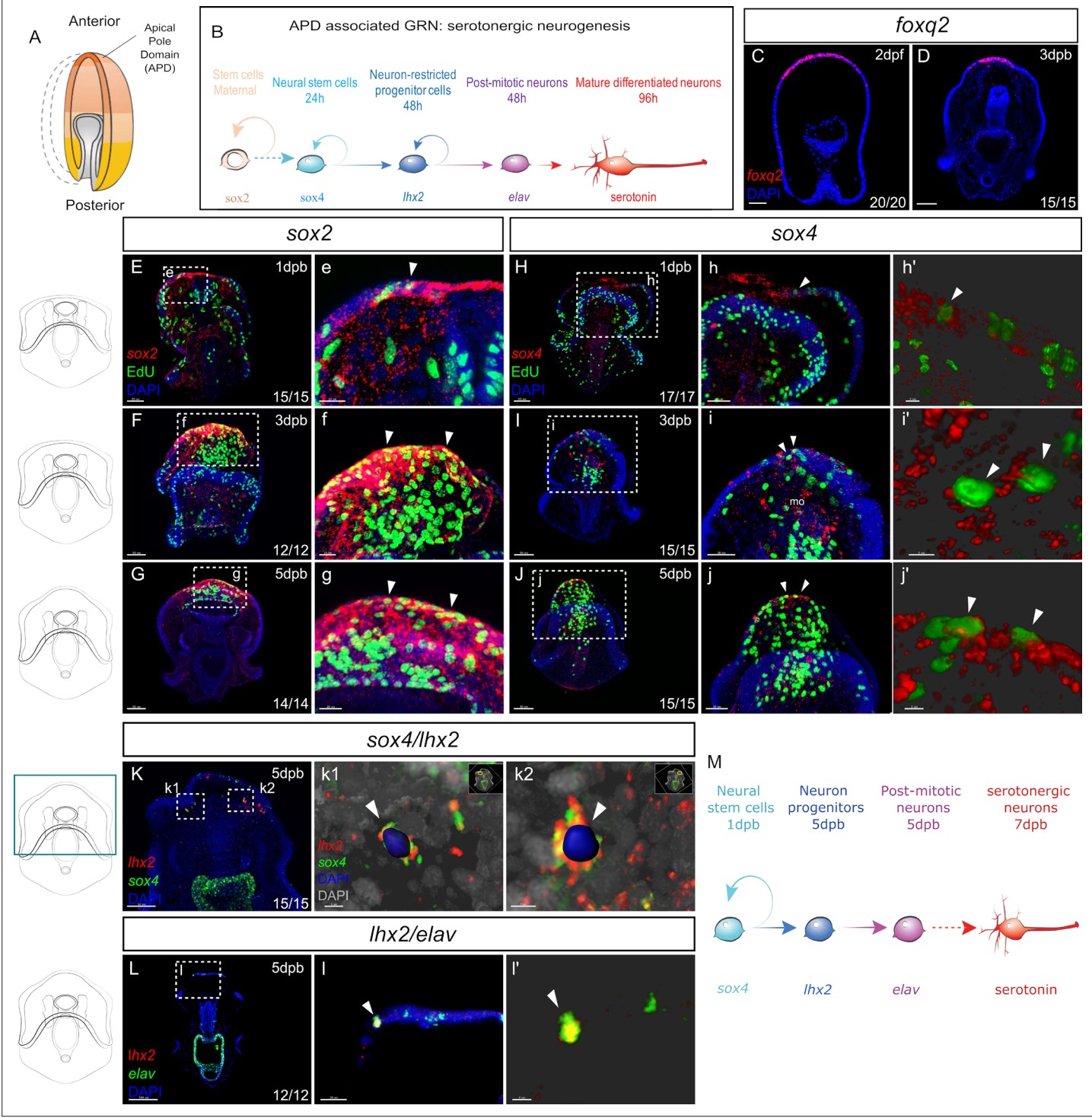

**Figure 2.** Recapitulation of embryonic neural gene expression during regeneration. (**A**) In normal development, the embryonic apical pole domain (APD) gives rise to serotonergic neurons via the APD-associated neurogenesis pathway illustrated in (**B**). (**C–D**) Fluorescent in situ hybridization (FISH) of APD marker gene *foxq2* in embryos (**C**) and in the leading edge of the regenerating larvae at three dpb (**D**). (**E-j'**) EdU labeled and FISH (EdU-FISH) results show that *sox2* and *sox4* are expressed in proliferative cells at the regeneration leading edge. (**E–G**) EdU-FISH of gene *sox2*. (**e–g**). Expanded view of the boxed area in (**E–G**). (**E**) Upon bisection, *sox2+* cells are highly concentrated at the wound site and later in (**F–G**) at the regeneration leading edge. Throughout the time-course, *sox2+* cells constantly undergo cell cycling (**e–g**). (**H–J**) EdU-FISH of *sox4*. (**h–j**). Expanded view of the boxed area in (**H–J**). (**H**). Upon bisection, *sox4+* cells are detected at the regeneration leading edge and later in (**I–J**) expanded to the mouth. Throughout the time-course, *sox4+* cells constantly undergo cell division (**h–j**). (**h'-j'**) 3D visualization of EdU + *sox4+* cells using Imaris software, highlighted in (**h–j**), indicated

*Figure 2 continued on next page*

*Figure 2 continued*

by white arrowheads. This shows clear double detection of EdU and *sox4*: the EdU + nucleus is surrounded by *sox4* signal. (**K-i′**) Double fluorescent in situ hybridization (FISH) shows the recapitulation of APD gene expression trajectory. (**K**) In 5dpb larvae, *sox4* and *lhx2* are co-expressed in cells at the lateral regeneration leading edge. (**k1′-k2′**) show the Imaris 3D reconstructed view of boxed areas k1 and k2. In the highlighted cells (arrowheads), nucleus labeled with DAPI is surrounded by both *lhx2* and *sox4* signals, indicating co-expression in the same cell. Scale bar in (**c1**): 5 μm; (**c2**): 7 μm. (**L**) In 5 dpb larvae, *lhx2* and *elav* are co-expressed in cells at the lateral regeneration leading edge, scale bar: 100 μm. This is amplified in (**l**). (**l′**) shows the 3D reconstructed view of the cell marked by arrowheads in (**l**) (**M**) A proposed Model for the regeneration of neurons in sea star larvae. Dpb: day-post-bisection. Scale bar in (**E–K**): 50 μm; (**L**): 100 μm; (**e–i,l**): 20 μm; (**j**): 30 μm; (**h′-j′,k1 and l′**): 5 μm; (**k2**): 7 μm. Dpf: day-post fertilization; dpb: day-post bisection; mo, mouth. The numbers shown in the lower right corner of each image indicate the number of larvae showing a positive FISH signal among the larvae examined.

The online version of this article includes the following figure supplement(s) for figure 2:

**Figure supplement 1.** Whole-mount in situ hybridization (WMISH) results show the reconstruction of the AP body axis.

**Figure supplement 2.** WMISH of embryonic neurogenic pathway genes during regeneration.

Collectively, these data show that bisection leads to the activation of embryonic neurogenesis gene expression states. *Sox2* and *sox4* expressions are induced upon wounding, and *sox4+* cells located at the lateral leading edge give rise to *lhx2+* cells. The *lhx2+* cells at the leading edge in turn become post-mitotic neurons expressing *elav* (***Figure 2M***). The location of *lhx2* expression at the leading edge, within the zone of *foxq2* expression, also recapitulates the spatial localization of embryonic serotonergic cell specification.

## Newly specified sox4+ neural cell lineage is induced by bisection

The gene expression states propose the intriguing model that neural progenitors, that is newly specified *sox4+* cells, arise at the leading edge following bisection, and that these newly specified neural progenitors will reform the serotonergic neurons by entering the embryonic neurogenesis pathways. To test this hypothesis, we needed to establish a reporter system that would distinguish new *sox4+* cells from existing *sox4+* cells and also follow their fate for several days.

To this end, we developed a highly stable, photoconvertible fluorescent reporter. We modified an approach we have used previously in which eGFP (GFP) was homologously recombined into the *sox4* coding region of a Bacterial Artificial Chromosome (BAC) clone, to replace the coding sequence of sox4 with that of GFP while leaving the endogenous basal promoter and 5′UTR in place (***Buckley and Ettensohn, 2019***). When the sox4:GFP recombinant BAC is injected into embryos, it faithfully expresses GFP in *sox4+* cells, recapitulating endogenous *sox4* expression (***Cheatle Jarvela et al., 2016***). It should be noted that transgenic DNAs are inherited mosaically and thus any individual larva expresses the reporter in only a subset of cells. This demonstrates that the approximately 150 Kbp surrounding the genomic locus of the *sox4* coding region, including the endogenous basal promoter, is sufficient to drive correct expression of the transgenic GFP reporter gene.

We now exchanged GFP for the photoconvertible Kaede sequence. Kaede fluorescent protein (FPbase ID: Q31FH) stably converts from green to red after exposure to UV light (***Ando et al., 2002***). We experimentally measured the stability of Kaede, by injecting in vitro synthesized kaede mRNA into zygotes and quantifying fluorescence over time. We show that translated Kaede protein is detectable above background for at least 7 days in *P. miniata* embryos and larvae when grown at 15°C (***Figure 3—figure supplement 1***, and ***Figure 3—source data 1***). This supports previous work which shows that GFP is extremely stable in these embryos, likely because they are grown at low temperatures (***Arnone et al., 1997***).

The *sox4*:Kaede BAC construct (***Figure 3A***) was injected into zygotes, and larvae grown to 7dpf bipinnaria larval stage. The BAC DNA construct is stably inherited by clones of cells and therefore present mosaically in the cells of the larvae. The larvae were then bisected and immediately photoconverted by exposure to 405 nm light (***Figure 3—figure supplement 1***, and ***Figure 3—source data 1***). Larvae were inspected under fluorescent microscopy to assess photoconversion, and we confirm that all cells that had expressed *sox4* within the previous 7 days now fluoresce at 560 nm (red). Decapitated larvae were then individually cultured in 24-well dishes so that we could follow individual cell populations in each larva (***Figure 3C***). At 2dpb and 3dpb we observe green-only, red-only, as well as yellow (green plus red) cells (***Figure 3B and D–d***). Red cells in the regenerating larvae are the historically labeled cells that have expressed *sox4* at some point prior to bisection but no longer

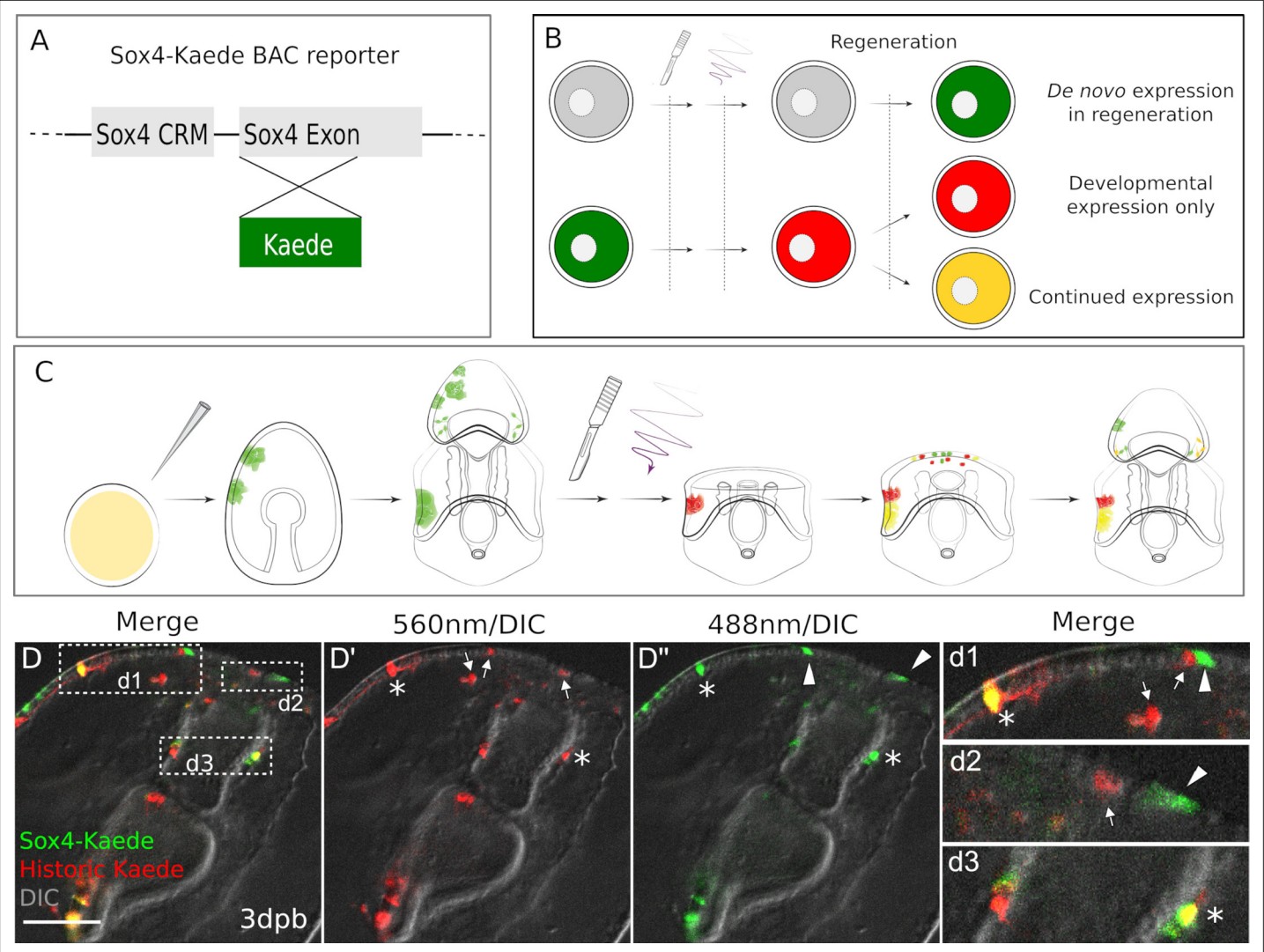

**Figure 3.** Existing and de novo Sox4+ lineages regenerate serotonergic neurons. (**A**) The design of Sox4:Kaede BAC construct. (**B–C**) Design of photo-conversion experiment. Sea star zygotes are injected with the Sox4-Kaede BAC. Injected embryos are incubated till larvae for bisection. Photo-conversion is immediately performed on bisected posterior segments. After regenerating for periods of time as shown (dpb), larvae are observed to examine the Sox4+ lineages. (**D**) In Sox4-Kaede transgenic regenerating larvae, there are multiple sources of Sox4+ cells at the regeneration leading edge. Some Sox4+ cells are derived from the yellow, existing Sox4 lineage (marked by asterisks). Some Sox4+ cells are differentiated cells that no longer express Kaede (arrows). There are also green Sox4+ cells that are newly specified upon decapitation (arrowheads). The boxed areas in the regeneration leading edge are amplified in (**d1**), (**d2**) and (**d3**). Scale bar in (**D**): 50 μm. At least 30 regenerating larvae in two independent batches were examined.

The online version of this article includes the following source data and figure supplement(s) for figure 3:

**Source data 1.** Quantification of Kaede colors and protein stability.

**Figure supplement 1.** Stability of Kaede protein in P.miniata during development.

express *sox4*. For example, these may have been differentiated neurons that no longer express the progenitor marker *sox4*. The yellow cells are from existing larval and regenerative *sox4*+ cell lineages. That is, these cells expressed *sox4* in the intact larvae and continue to express *sox4* in their progeny after wounding. Most interestingly, there are green-only cells. These are cells that have not previously expressed *sox4* but are induced to start expressing *sox4* only upon decapitation.

We questioned whether the presence of newly expressing *sox4*+ cells is induced by wounding, or whether there is normally a pool of progenitor cells that are set to become *sox4*+ over the timepoints we observed. We therefore compared the rate of *sox4*+ cell specification between embryogenesis and

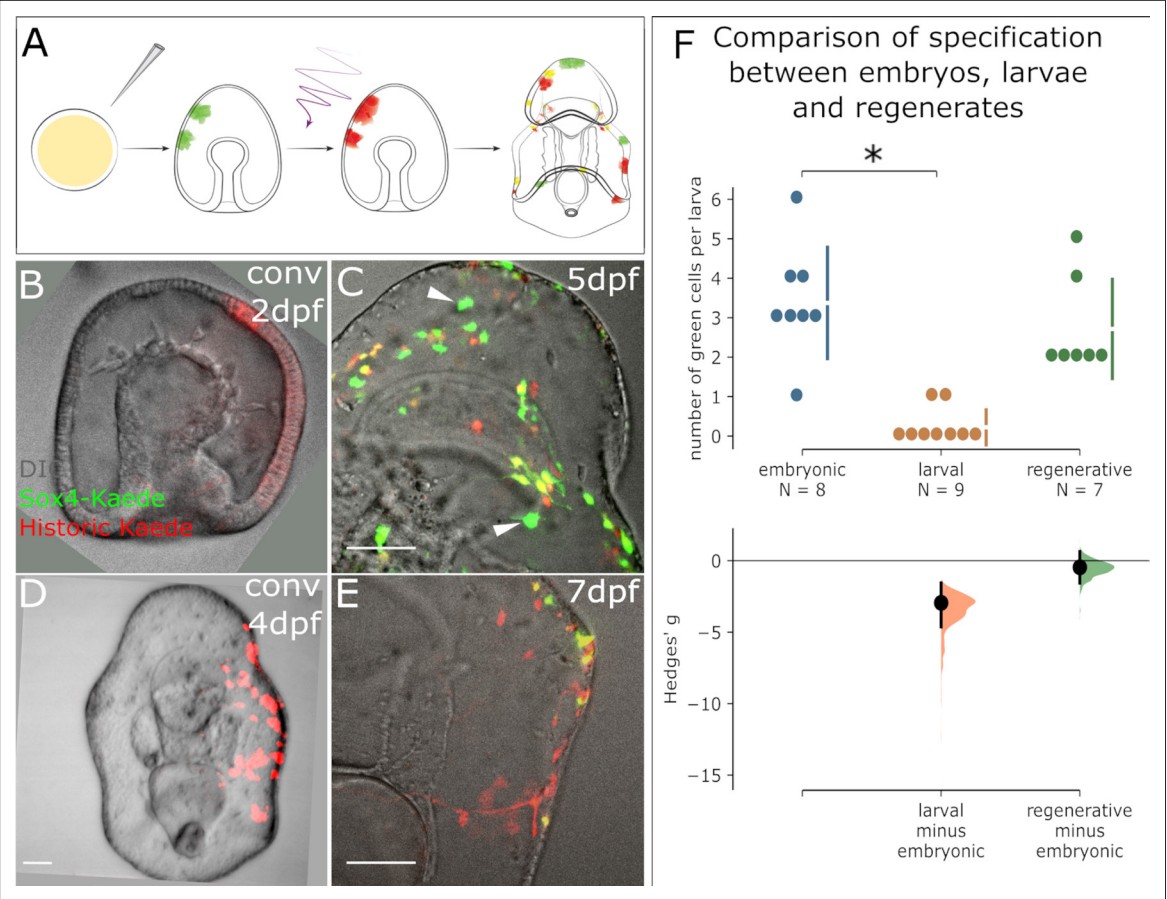

**Figure 4.** Appearance of new Sox4+ cells ends at the larval stage but resumes in regeneration. (**A**) A schematic for the experimental plan to determine the source of normal larval Sox4+ cell lineage. Sox4:Kaede transgenic embryos were photoconverted at either embryonic stage or larval stage. Sox4 expression was examined three days later. (**B**) Embryo injected with Sox4-Kaede BAC is converted at 2dpf. Red Kaede localization therefore marks all labeled Sox4+ cells. (**C**) By 5dpf there are several newly specified, green Sox4+ cells indicated by arrowheads. However, (**D**) when converted at 4dpf (larval stage), only yellow cells and red cells are observed 3 days later as shown in (**E**), indicating no specification of Sox4+ cell occurs in the larva. (**F**) Quantification of Sox4 specification in embryos, larvae and regeneration. The Hedges' g for two comparisons against the embryonic group are shown in the above Cumming estimation plot. The raw data is plotted on the upper axes. On the lower axes, mean differences are plotted as bootstrap sampling distributions. Each mean difference is depicted as a dot. Each 95% confidence interval is indicated by the ends of the vertical error bars. Sox4+ cell specification event is close to 0 by 4dpf, significantly decreased compared to the embryonic (2dpf) state. Specification of Sox4+ cells is restored in regenerating larvae. Scale bar in (**B–E**): 50 µm. dpf: day-post-fertilization. At least 40 embryos were examined from five independent batches.

The online version of this article includes the following source data for figure 4:

**Source data 1.** Quantification of *sox4* green cells.

larval development. We again injected Sox4:Kaede BAC into zygotes and photoconverted either 2dpf embryos or 4dpf larvae (*Figure 4A*). We then quantified the numbers of green, red and yellow cells 3 days later (5dpf or 7dpf) over normal development (i.e. without bisection, *Figure 4—source data 1*). We find that there are high numbers of green+ cells in larvae photoconverted at 2dpf (*Figure 4B–C*), but almost none in larvae converted at 4dpf (*Figure 4D–E*).

This is evidence that new *sox4*+ cells are actively being specified in embryos, but that this specification has stopped by 4dpf (i.e. larval stage). Comparing the numbers of green (newly expressing) *sox4*+ cells during normal development with those counted in regeneration, we see that specification of new *sox4*+ cells is highest in embryos, ends by late embryogenesis, and resumes in response to bisection induced regeneration (*Figure 4F*, and *Figure 4—source data 1*). Thus, decapitation induces a wound response that re-initiates the specification of new *sox4*+ cells, a process which had stopped by day four of embryogenesis. These *sox4*+ cells then reenter into embryonic neurogenesis pathways (*Figure 2*).

## Sox2+ progenitor cells are specified to Sox4+ cells to regenerate neurons

We have shown that there arises a new population of sox4+ cells induced by bisection. During normal embryogenesis, the neural precursor cells are derived from *sox2+* ectoderm. *Sox2* is expressed throughout the ectoderm in embryos and therefore broadly marks this germ layer (*Yankura et al., 2013*; *Yankura et al., 2010*). We therefore asked whether these new bisection-induced neural progenitors similarly arise from ectodermal lineages.

Our gene expression analysis showed that *sox2* is broadly upregulated throughout the anterior of the regenerating larva (*Figure 2E–g*) and that new, dividing, *sox4+* cells also form in this region (*Figures 2H–j and 3D*). We therefore generated a sox2:Kaede BAC. We first tested whether this BAC recombinant recapitulated normal Sox2 expression in sea stars. After BAC injection, zygotes started to express Kaede at 24-48hpf in broad patches of ectodermal cells. By larval stage, Kaede was detected on the ectoderm including the ciliary bands, dorsal ganglia, and ectodermally derived neurons (*Figure 5—figure supplement 1A-C'*). This localization pattern recapitulates the embryonic *sox2* expression pattern reported previously (*Yankura et al., 2013*; *Yankura et al., 2010*; *Figure 2—figure supplement 2A*).

We next generated a sox4:Cardinal BAC recombinant so that we could examine the potential colocalization of sox2+ and sox4+ lineages. This is the same BAC recombinant as sox4:Kaede (*Figure 3A*) but with the Kaede coding sequencing replaced by Cardinal (FPbase ID: 7EWEM), which fluoresces in the far-red range (633 nm). These two BAC recombinants were co-injected to identify potential double labeling of *sox2+* cell lineages (green, red, or yellow depending on the assay) and *sox4+* cell lineages. It is important to note that multiple constructs when co-injected into echinoderm zygotes are then co-integrated, and are therefore inherited in the same cellular lineages (*Cheatle Jarvela et al., 2014*). When we examine expression in normal developing larvae, we find that sox4+ cells overlap completely with *sox2+* cells on the ectoderm as is expected in normally developing larvae (*Figure 5—figure supplement 1D-D''*), although not all *sox2+* cells are *sox4+*, as *sox2+* also produce non neural ectoderm.

We next photo-converted the double BAC recombinant larvae (i.e. all sox2:Kaede+ cells are converted to red), and immediately bisected the larvae and allowed them to regenerate for 2–3 days (*Figure 5A*). Strikingly, we found new (green+ cells) expression of *sox2*, indicating that cell that had not previously expressed *sox2* are now expressing the *sox2* gene following bisection. Furthermore, four of 15 observed larvae exhibited cells with colocalized green Kaede (new *sox2* expression) and Cardinal (*sox4*,far red, pseudo-colored blue) at the regenerating leading edge (*Figure 5 B(b-b3)*)at 3dpb. For example, in the larva shown in *Figure 5*, there are 3 cells located at the left side of the regeneration leading edge that contain green Kaede and Cardinal. As the green Kaede reporter labels de novo sox2 expression, this shows that these new *sox2+* cells start to express *sox4* and thereby enter into neurogenesis pathways. We observe only few newly expressing *sox2+* and *sox4+* cells in this assay, likely as our reporter system is mosaic, and because there are in reality only small numbers of newly expressing *sox2+* cells at this stage. Our whole mount in situ analyses in *Figure 2* also suggested that there will be new *sox2+* cells, and that these will become sox4+. The Cardinal reporter can potentially represent both pre-existing and regenerative *sox4* expressions. However, it is important to note that at the time of bisection, all labeled sox4+ cells overlapped with *sox2+* lineage in the imaged larvae. In other words, all Cardinal + cells are also red Kaede+ at the time of photoconversion. Thus, the sox4:Cardinal in *Figure 5B* must be expressed after wounding. This therefore supports the hypothesis that cells derived from non *sox2+* lineages are induced to express *sox2* in regenerating larvae and will contribute to neural progenitors. Indeed when we follow these larvae through to 7dpb (*Figure 5—figure supplement 2A-B*) we find fluorescence in the neural projections which are characteristic of fully formed neurons (*Figure 5—figure supplement 2a1-a3*).

## Bisection induced Sox2+ cells maintain usual embryonic cell restriction

We were interested to determine whether these newly specified sox2+ cells, which were not previously ectodermal, now maintain an ectodermal lineage. We examined at least 10 larvae over three independent injection batches and found that in all regenerating larvae the newly expressing *sox2+* cells contributed to only tissues normally derived from sox2+ lineages (e.g. *Figure 5—figure supplement 1E-F''*). We observed sox2:Kaede green cells in the outer ectodermal epithelium, the mouth,

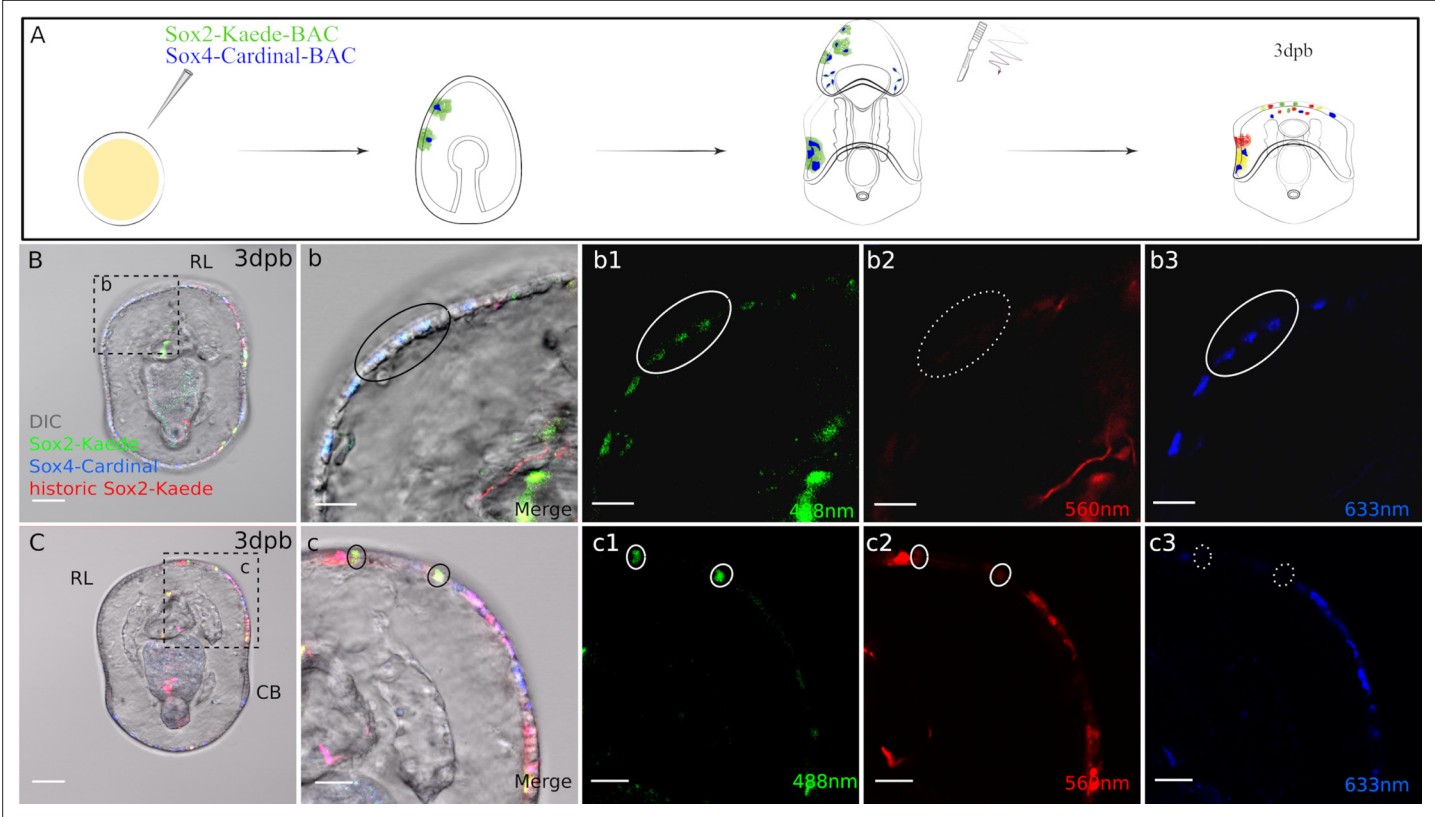

**Figure 5.** Sox4+ cells are specified from Sox2+ cells at the regeneration leading edge. (**A**) Schematic representation of the experimental design. Two BACs: sox2:Kaede and sox4:Cardinal were coinjected. Seven days old transgenic larvae were bisected, Kaede protein was photoconverted immediately following bisection, then regenerated larvae were analysed at 3 days later. (**B3**) Some sox4+ cells at the regeneration leading edge are made from de novo sox2+ cells. Boxed area in (**B**) is amplified in (**b**). In 3dpb transgenic larvae, sox4+ cells with de novo sox2 expression are located at the lateral side of the regeneration leading edge (black solid circle). These cells have (**b1**) newly expressed, regenerative sox2:Kaede and (**b3**) sox4:Cardinal expression (both highlighted with white solid circle), (**b2**) but have no historic sox2:Kaede (white dotted circle). (**C**). Sox2+ cells at the regeneration leading edge have multiple sources. (**c**) Is the amplified view of the boxed area. Apart from the de novo sox2+ cells, there are also yellow, larval sox2 lineage at the regeneration leading edge (black solid circle). These cells do not enter sox4-mediated pathways. They are labeled with (**c1**) green and (**c2**) red Kaede (white solid circle), (**c3**) but not Cardinal (white dotted circle). Images B and C are representative of at least 15 observed samples. Scale bar in (**B–C**): 50 µm; (**b–b3, c–c3**): 20 µm; (**d-d3**). RL, regeneration leading edge; CB, ciliary band. Dpb: day-post-bisection. Sox2: Kaede (red, yellow, green) and Sox4: Cardinal (far red, false-colored to blue).

The online version of this article includes the following figure supplement(s) for figure 5:

**Figure supplement 1.** Expression pattern of sox2-Kaede BAC.

**Figure supplement 2.** Sox2+ cell lineage forms the regenerated nervous system.

and ciliary bands. This indicates that once induced these cells will maintain their normal, embryonic fate potential.

It is worth noting that not all sox2+ cells enter the neurogenesis pathway in regeneration (*Figure 5* C(c1-c3)). For example, in the larva shown in *Figure 5*, there are two cells at the regeneration leading edge that contain both green and red Kaede, derived from the sox2 lineage. These cells do not express sox4:Cardinal by 3dpb. This suggests that while larval sox2 cell lineages are restricted to ectodermal potency, they do not necessarily contribute only sox4+ neural fates.

## New sox2 expression is induced in the absence of cell division

Finally, we questioned whether new *sox2* expression could arise in the absence of cell division. Stem cells, by definition, require an asymmetric division to produce a new daughter cell type and maintained stem cell. Thus new *sox2* expression should not be present when cell division is blocked if it is solely derived from asymmetric stem cell division. We bathed sea star larvae in Aphidicolin, a conserved inhibitor of cell division, that is particularly well characterized in echinoderms where it is

known to block DNA polymerase function, nuclear break down and microtubule organizing centers (*Mashanov et al., 2017*; *Nishioka et al., 1984*). We first used a serial dilution test to determine that the 25 µM of Aphidicolin was the minimum concentration needed to reliably inhibit cell division as assayed by EdU staining, while cells in DMSO controls divided normally (*Figure 6B*). We incubated regenerated larvae in the drug or vehicle for up to 1 week, with daily changes into fresh drug or vehicle, and analyzed them every 24 hr to test that Aphidicolin consistently blocked cell division over this time (*Figure 6—figure supplement 1*). We then microinjected the *sox2*:Kaede recombinant into fertilized oocytes, bisected 7dpf larvae, photoconverted and added Aphidicolin or DMSO as a control in the sea water immediately after post bisection. Bisected transgenic larvae were continuously incubated in drug or control solutions over three days (*Figure 6A*). The aphidicolin treatment, therefore, covered the phases of regeneration, and preceded the first observed *sox2* and *sox4* expression by at least one day. At least 10 larvae over three independent injections were analyzed. We then assayed for red, green, or yellow-positive cells (*Figure 6C–D"*, and *Figure 6—source data 1*). We show that there are multiple yellow, green cells along the leading regenerated edge. This confirms that cells which were not previously expressing *sox2* are now able to initiate expression of this gene in the absence of cell division and are thus not the result of asymmetric stem cell division.

This experiment additionally addresses a concern that green FP only cells may actually be Green + Red, but have lost red FP signal through dilution if the precursor cells are highly proliferative. We thought this unlikely as green+ FP cells are identified prior to extensive regenerative proliferation and we individually followed larvae to track previously identified Red FP cells. We now however, can definitely exclude dilution based on proliferation as a cause for loss of Red FP as Green FP cells emerge in the absence of any division.

## Discussion

The ability that some animals have to restore tissues and body parts following traumatic injury challenges our current understanding of the mechanisms of cell specification and differentiation. Significant questions remain surrounding the source of new cells needed to regenerate lost tissues and organs, their potencies and whether developmental GRNs are re-used. A further open question is whether any of these processes are shared across taxa, and can explain the differing capacities for regeneration.

Echinoderms are an especially important system for the study of these questions as they are well known for their extensive capacities for regeneration (*Ben Khadra et al., 2018*; *Byrne, 2020*). There has, therefore, been extensive research on regeneration of multiple species, tissues and stages in echinoderms (*Vickery et al., 2001*; *Kasahara et al., 2019*; *García-Arrarás et al., 2018*; *Mashanov and Zueva, 2019*; *Piovani et al., 2021*). While providing important insight and an emerging consensus on the role of local dedifferentiation, the availability of transgenic tools has precluded the cell lineage analysis needed to definitively identify cell sources and their potencies. The discovery that many echinoderm larvae are able to regenerate (*Wolff and Hinman, 2021*), opens the possibility to use genetic engineering tools that are available for embryos, to probe regeneration. Additionally, developmental GRNs and the cell specification pathways are extremely well characterized for many species of echinoderms, providing an opportunity to compare developmental to regenerative GRNs of the same larval cell types. In this study, we develop a new BAC-based transgenesis protocol to identify and track the cellular sources of regenerated neurons of the sea star bipinnaria larva, and to compare neurogenesis pathways between development and regeneration.

We show that bipinnaria larvae of the bat star, *P. miniata*, are able to regenerate their anterior nervous system. Sea star larvae have remarkably complex nervous systems (*Carter et al., 2021*; *Elia et al., 2009*) given their small size and presumably simple behavior. We took advantage of the highly specific marker of differentiated serotonergic neurons, which includes immunoreactivity to anti-serotonin antibody (Sigma, S5545), characteristic axonal projections, and location within the larvae, to confirm that these neurons are removed by surgery and reform within 21 days. The removed pre-oral ciliary band and lip of the mouth are also reformed during regeneration.

We used fluorescent in situ hybridization (FISH) to show that genes involved in embryonic AP axis specification are re- expressed along the axis of bisected larvae. For example, *foxq2*, *six3,* and *wnt3* are key nodes in the embryonic GRN for axial specification (*Cheatle Jarvela et al., 2016*; *Yankura et al., 2013*), and are re-expressed along the AP axis of the posterior regenerating segment. Reestablishing

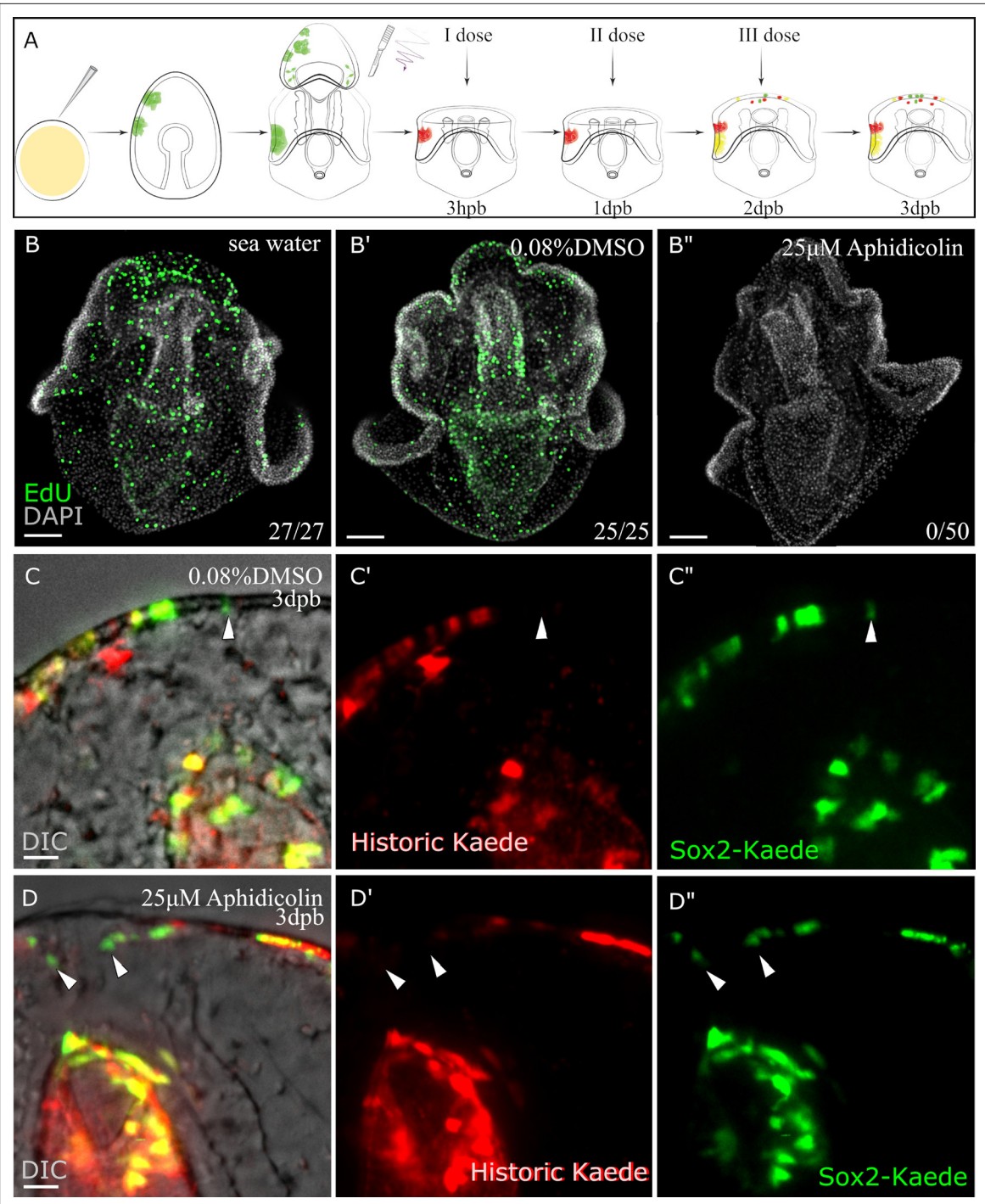

**Figure 6.** Sox2+ cells fate specification (continued or de novo expression) does not depend on entering into the cell cycle during the early stage (upto 3dpb) of regeneration. (**A**) Schematic representation of the experimental design. Sox2:Kaede transgenic larvae were bisected and photoconverted. The first dose of 25 µM aphidicolin or vehicle (0.08% DMSO) were applied after 3hpb, changed every 24 hr until 3dpb then the regenerated larvae were analyzed. (**B-B″**) To monitor the effect of aphidicolin on DNA synthesis, we used an EdU cell proliferation assay. 3dpb larvae in aphidicolin, DMSO or sea water solutions were incubated in 15 µM EdU for 1 hr. (**C–D**) Sox2-Kaede cell specification by three dpb under aphidicolin (**D-D″**) or vehicle (**C-C″**) treatment. Scale bar in (**B-B″**), (**C-D**) is 50 µm. Dpb: day-post-bisection. Hpb: hour-post-bisection. Arrowheads highlight de novo sox2 expression. The numbers shown in the lower right corner of (**B-B″**) indicate the number of larvae analyzed. (**C** and **D**) images are representative of at least 15 transgenic samples from two independent batches. A percentage of yellow (re-used) and green (de novo) Kaede expressing cells at the wound site during vehicle or drug admission are 71.6 and 2.1; 80 and 1.7 correspondent (***Figure 6—source data 1***).

The online version of this article includes the following source data and figure supplement(s) for figure 6:

*Figure 6 continued on next page*

*Figure 6 continued*

**Source data 1.** Quantification of *sox2* cells under drug/vehicle treatment.

**Figure supplement 1.** Aphidicolin or vehicle treated bisected larvae during one week of regeneration.

axial patterning is a commonly observed early feature in many species undergoing whole body regeneration (*Gehrke et al., 2019*; *Reddien, 2019*), which we now show also occurs in sea star larvae. *Sox2* and *sox4* expression is similarly induced adjacent to the wound site within one day. This region will form a proliferative epithelial zone by the third day following bisection. In embryos, *sox2* is expressed throughout the ectoderm, while *sox4+* cells are scattered throughout the ectoderm where they function as precursors for multiple neural types in the larvae (*Cheatle Jarvela et al., 2016*). This is the first indication that there is a resetting of gene expression states adjacent to the wound and is consistent with a broad respecification of these cells.

One of the most fascinating questions in regeneration is whether animals reuse their embryonic GRNs to regenerate cell types. If different pathways are used, this indicates that there are regeneration-specific GRNs deployed after traumatic wounding. This further suggests that such pathways may independently evolve under different selection pathways than the embryonic pathways. If embryonic GRNS are reused, the challenge is to understand how they are reactivated in non-zygotic cells. The new expression of embryonic axial patterning genes across the bisected larva (*Figure 2C–D*) is consistent with re-entry into an embryonic state. We further show, using FISH and using IMARIS to confidently assess colocalization of signal, that *sox2+* cells overlap with *sox4+* cells, that *sox4* expression overlaps with *lhx2* expression, which in turn overlaps with *elav* (*Figure 2*). We have previously demonstrated that these gene products mark the embryonic neurogenesis pathways, and that ectodermal *elav* expression is a marker of differentiated neurons (*Cheatle Jarvela et al., 2016*). We have also previously shown that *lhx2* is needed for the differentiation of serotonergic neurons (*Cheatle Jarvela et al., 2016*). This data therefore supports the hypothesis that embryonic serotonergic neurogenesis pathways are reactivated in the induced *sox2+* cells, although we are not yet able to show that any of these gene products are functionally required for neurogenesis in the regeneration context.

The presence of *sox2+* and *sox4+* cells adjacent to the wound site provide the possibility that these are the cellular source of regenerated neurons. Cell tracking however is needed to test whether these cells do indeed form neurons, and to identify the cellular origin of *sox2* and *sox4* expression. The wound adjacent expression could arise from existing *sox2+* or *sox4+* cells in the larvae. This would suggest that these cells function as a resident stem cell population of regenerated neurons. Alternatively, they would arise from other, existing, cells, which may either be derived from an additional source of stem cell or through de- or trans-differentiation. We therefore developed a novel transgenic reporter system based on BAC recombineering that takes advantage of highly stable, irreversibly photoconvertible, fluorescence Kaede reporter (*Ando et al., 2002*). The injected BAC DNA construct is stably inherited in clonal lineages of cells after one to several embryonic cell division and hence the reporter will express mosaically. In the system we developed, cells that are currently expressing the gene of interest will fluoresce green, and may also contain red FP if they have previously expressed the gene prior to photoconversion. Using this reporter system, we show that the *sox4* gene is expressed in existing *sox4+* cells, and in cells that have not expressed *sox4* within the last seven days (*Figure 3*). We reasoned that new neural precursor cells (i.e. green only *sox4+* cells) may be continuously generated as a normal development process and therefore not specifically induced by wounding. We therefore compared the formation of new *sox4+* neural precursors in development and regeneration. We find many new sox4+ cells being produced in early embryos (days 2–5 development) as expected, but this has almost entirely stopped by swimming larval phase (days 4–7 development) and resumed upon wounding (*Figure 4*). The subsidence of *sox4+* cell formation in the larval stage is consistent with data showing that the serotonergic postmitotic neuronal number is stable in larvae from day 4 or 5 of development (*Figure 1—figure supplement 2*). Using the same experimental strategy, but now with sox2 BAC reporter, we show the *sox2* gene is also expressed in existing and new cells. Using an additional Cardinal reporter, we show that some, but not all, of these new *sox2+* cells become *sox4+* (*Figure 5*), indicating that they can become neural precursors but also other cell types. We followed these reporters through regeneration to show that they are later found in cells with characteristic axonal projections and located in the lower lip of the mouth, and dorso-lateral domains adjacent to the mouth as stereotypic of

serotonergic neurons. Thus, we show that these wound-induced *sox2* cells do become neurons, other ectodermal cells and mouth endo/ectoderm in a similar lineage potential as the embryonic *sox2*+ lineage.

The existing, *sox2*+ *sox4*+ cells that continue to express these genes (i.e. the yellow cells in our assay) likely function as resident ectodermal- and neural- stem cells as we show that they are dividing and contribute to neural and ectodermal tissues. We wanted to investigate whether the new *sox2*+ cells (green) arose from existing cells or might arise from another, possibly totipotent resident stem cell population. As stem cells must, by definition, divide to produce a daughter that can be directed along a new differentiation pathway we used small molecule inhibitors to block cell division following bisection. We find that new *sox2*+ cells still emerge in these larvae (*Figure 6*). This therefore shows that new sox2+ expression does not require the asymmetric division from a stem cell progenitor, and instead, therefore, is the result of respecifying existing cells.

The human and mouse orthologs of the sea star sox2 gene have well-characterized functions in the induction of pluripotency and maintenance of stem cells, especially neurogenic stem cells (*Feng and Wen, 2015*). In combination with Oct4, Klf4, and c-Myc, sox2 can reprogram human and mouse somatic cells to pluripotent states (*Soufi et al., 2015*; *Takahashi and Yamanaka, 2006*). Other models of echinoderm regeneration have implicated, although not demonstrated, a role for orthologs of these pluripotency factors in de-differentiation (*Mashanov and Zueva, 2019*; *Mashanov et al., 2015a*). We have also previously shown that *Klf4* is expressed at the wound edge following bisection and is thus presumably co-localized with *sox2* (*Cary et al., 2019b*). The new wound-induced expression of *sox2*, therefore, presents the hypothesis that induction of *sox2* expression is the actual cause of the dedifferentiation of cells, and/or may now function to maintain an ectodermal stem cell population. We have not yet formally shown whether new induction of *sox2* leads to dedifferentiation (i.e. reprogramming into a progenitor state), or whether there are other types of reprogramming, as we do not know the differentiation state of cells induced to express *sox2* or whether these cells now adopt an identical molecular signature to existing *sox2* cells. The extent of lineage conversion are unknown and open questions in stem cell biology in any system, and yet is critical knowledge for translational applications of stem cells. The work here provides foundational knowledge in a tractable system to work toward answers to these questions. We have shown that once respecified, *sox2*+ cells are restricted to become ectoderm, and the lip of mouth ectoderm/endoderm cell types, indicating that they are multi-potent but germline restricted. In embryos, *sox2*+ cells are also restricted to ectodermal lineages and the lip of the mouth, and therefore, the wound-induced sox2+ cells now enter their normal, embryonic lineage. A subset of these induced cells recapitulate an embryonic neurogenesis pathway to reform serotonergic neurons around the mouth, and the latero-dorsal ganglia.

As new model systems are added to the study of regeneration, they present an opportunity to re-examine our understanding of regeneration and the ways in which cellular reprogramming is induced by wounding. Early concepts around cellular origins of whole body regeneration suggested a dichotomy between use of stem cells and de- or trans-differentiation, although more recent studies suggest a greater complexity, with a range of mechanisms often existing in any given species. For example, the freshwater planarian, *Schmidtea mediterranea*, utilizes a population of heterogeneous pluripotent somatic stem cells, called neoblasts, to proliferate and differentiate to replace body parts (*Sánchez Alvarado, 2006*). Recent work shows that specialized neoblasts can also divide to produce neoblasts that are pluripotent, suggesting a plasticity of their fate (*Raz et al., 2021*) Species such as Hydra and axolotl, refate differentiated cells either through dedifferentiation or transdifferentiation (*Gerber et al., 2018*). Although, Hydra also utilizes interstitial cells as a population of stem cells (*Siebert et al., 2008*). Cellular plasticity also varies widely, from the totipotent planarian neoblasts to lineage restricted progenitors used by vertebrate animals (*Kragl et al., 2009*; *Zhu and Pearson, 2016*).Our data indicate that echinoderm larvae induce non-stem cells into ectodermal neural stem cells, possibly through dedifferentiation or other mechanisms of fate conversion, as we see de novo sox2+ cells becoming neurons in regenerating larvae. This is reminiscent of the mechanisms of refating seen in cnidarian models such as *Hydra*. Additionally, we show that there is a maintained lineage of *sox2*+ and *sox4*+ progenitors in bisected larvae, also reforming the nervous system, as we see yellow neurons in regenerating larvae (*Figure 5—figure supplement 2A*). This at a broad level echoes the use of lineage restricted stem cells seen in vertebrates and other systems. We posit that echinoderms may have a natural capacity to maintain a cellular plasticity that permits induction of new lineage

stem-cells, as well as an ability to use existing lineage-restricted stem cells and that this flexibility may underlie their extensive abilities for regeneration.

# Materials and methods

## Key resources table

| Reagent type (species) or resource | Designation | Source or reference | Identifiers | Additional information |
|---|---|---|---|---|
| Commercial assay or kit | *Label* IT Nucleic Acid Labeling Kit, DNP | Mirus | MIR 3825 | |
| Commercial assay or kit | TSA Plus System | Perkin Elmer | NEL753001KT | |
| Commercial assay or kit | DIG RNA Labeling Kit (SP6/T7) | Roche | 11175025910 | |
| Commercial assay or kit | Click-iT Plus EdU Cell Proliferation Kit for Imaging, Alexa Fluor 488 dye | Invitrogen | C10637 | |
| Commercial assay or kit | mMESSAGE mMachine T7 transcription kit | Thermo-Fisher Scientific | AM1344 | |
| Antibody | Sheep polyclonal anti-digoxigenin AP-conjugate | Roche | 11093274910, RRID:AB_2734716 | IHC (1:2000) |
| Antibody | Sheep polyclonal anti-digoxigenin POD-conjugate | Roche | 1120773391, RRID:AB_514500 | IHC (1:2000) |
| Antibody | anti-DNP HRP-conjugate | Perkin Elmer | FP1129, RRID:AB_2629439 | IHC (1:500) |
| Antibody | Rabbit polyclonal anti-serotonin | Sigma | S5545, RRID:AB_477522 | IHC (1:250) |
| Antibody | Mouse monoclonal 1E11 | DSHB (*Nakajima et al., 2004*) | | IHC (1:5) |
| Antibody | Goat anti-mouse polyclonal Cy3 | Jackson Immuno-research | 115-165-146, RRID:AB_2491007 | IHC (1:2000) |
| Antibody | Goat anti-rabbit polyclonal Cy3 | Jackson Immuno-research | 115-165-144 | IHC (1:2000) |
| Antibody | Goat anti-rabbit polyclonal Alexa Fluor 488 | Invitrogen | A11008, RRID:AB_143165 | IHC (1:1000) |
| Recombinant DNA reagent | *sox2*:Kaede BAC | This paper | | http://echinobase.org |
| Recombinant DNA reagent | *sox4*:Kaede BAC | This paper | | http://echinobase.org |
| Recombinant DNA reagent | *sox4*:mCardinal BAC | This paper | | http://echinobase.org |
| Recombinant DNA reagent | *sox4*:GFP BAC | This paper | | http://echinobase.org |
| Chemical compound, drug | Aphidicolin | Sigma | A0781 | 25 µM |
| Software, algorithm | Fiji | Fiji (http://fiji.sc) | RRID:SCR_002285 | |
| Software, algorithm | Inkscape | Inkscape (https://inkscape.org/en/) | RRID:SCR_014479 | |
| Software, algorithm | GNU Image Manipulation Program | GIMP (http://www.gimp.org) | RRID:SCR_003182 | |
| Software, algorithm | Imaris | Imaris (http://www.bitplane.com/imaris/imaris) | RRID:SCR_007370 | |

## Culture and regeneration

Adult Sea star, *Patiria miniata*, were obtained from the southern coast of California, USA (Pete Halmay or Marinus Scientific) and housed in artificial sea water (ASW, Instant Ocean, Aquarium Systems) at 12°C–15°C. Embryos were cultured in ASW and fed Rhodomonas lens ad libitum as previously described (*Cheatle Jarvela and Hinman, 2014*). All studies of regenerating larvae were conducted

with larval cultures beginning at 7dpf at which point the larvae were manually bisected stereotypically through the foregut, midway along the AP body axis as described earlier (*Cary et al., 2019b*). Regenerating larvae were photographed using differential interference contrast (DIC) optics over the course of regeneration on a Leica DMI4000B microscope at ×100 magnification using Leica Application Suite software (Leica; Wetzlar, Germany).

## Whole-mount in situ hybridization (WMISH)

Embryos, intact and regenerating larvae for WMISH were fixed in 4% paraformaldehyde (PFA) in a high-salt MOPS-fix buffer (100 mM MOPS pH 7.5, 2 mM $MgSO_4$, 1 mM EGTA, and 0.8 M NaCl) for 90 mintes at room temperature or overnight at 4 °C. Following fixation, embryos were washed in (v/v) 25%, 50%, 75%, and 100% ice-cold 70% ethanol. Fixed embryos and larvae were stored at –20 °C. WMISH was conducted as described previously (*Hinman et al., 2003*; *Yankura et al., 2010*) with the following modification. Hybridization of sox4 and lhx2 riboprobes (final concentration of 0.2 ng/mL) was performed at 55 °C for 5 days. Hybridization of elav riboprobes (final concentration of 0.2 ng/mL) was performed at 58 °C for 5 days. Hybridization of foxq2, wnt3 and six3 riboprobes (final concentration of 0.1 ng/mL) was performed at 58 °C for 5 days. Larvae were imaged on a Leica DMI4000B inverted light microscope using DIC microscopy at ×100 magnification and the Leica Application Suite software (Leica; Wetzlar, Germany). Primer sequences used to prepare insitu probes are listed in *Supplementary file 1*.

## Double fluorescent in situ hybridization (FISH)

Intact and regenerating larvae of *P. miniata* at the indicated time points were fixed in a solution of 4% paraformaldehyde in MOPS-fix buffer (0.1 M MOPS pH 7.5, 2 mM $MgSO_4$, 1 mM EGTA, and 0.8 M NaCl) for 90 min at room temperature and transferred into 70% ethanol for long-term storage at –20 °C. Double FISH experiments were performed as previously described (*McCauley et al., 2013*) using digoxigenin-labeled antisense RNA probes (final concentration of 0.1–0.2 ng/mL) and dinitrophenol-labeled antisense RNA probes (final concentration of 0.1–0.2 ng/mL). Hybridized probes were detected using anti-DIG-POD antibody (1:1000, Roche, RRID:AB_514500), anti-DNP-HRP antibody (1:1000, Perkin Elmer, RRID:AB_2629439) and tyramide signal amplification (Perkin Elmer). A 1:100 dilution of Cy3 or FITC labeled tyramide in an amplification buffer was used to treat larvae for 7 min at room temperature in the dark. A Cy3- or FITC-labeled tyramide was deposited near the hybridized probe in a horseradish peroxidase mediated reaction. This allowed for fluorescence detection of labeled probes. During PBST washes, larvae were incubated for a total of 20 min in solution with 1:10,000 dilution of 10.9 mM DAPI, followed by PBST washes. Larvae were photographed with Zeiss 880 Laser Scanning Microscope at ×200 magnification with 405 nm, 488 nm, and 560 nm channels in Z-stack settings.

## EdU-FISH

Labeling and detection of proliferating cells in *P. miniata* intact and regenerating larvae were performed using the Click-it Plus EdU 488 Imaging Kit (Life Technologies), with the following modifications. Larvae were incubated in a 10 µM solution of EdU for 15 min or 6 hr in seawater at 15 °C followed by immediate fixation in 4% PFA in PBS buffer with 0.1% Triton x-100. Fixation was performed at room temperature for 90 min. FISH was conducted as described above in the double FISH method with the following modification. Either FITC- or Cy3-labeled tyramide was used to detect labeled probes.

## Immuno-fluorescent (IF) staining

Whole mount IF staining was performed as described elsewhere (*Cheatle Jarvela et al., 2016*). Briefly animals were fixed in 4% PFA prepared in PBS (pH 7.4) at room temperature for 15 min. The samples were post-fixed in ice-cold methanol for 10 min, allowing for setting of larvae by gravity. The samples were then washed in PBS, permeabilized in PBS/0.5% Triton X-100 for 30 min, then incubated in 0.1 M glycine for 30 min to quench autofluorescence. After another wash in PBS/0.1% Triton X-100 (3 × 15 min), the larvae were blocked using 3% BSA/PBS/0.1% Triton X-100 for 1 hr. The primary antibodies (*Supplementary file 2*) were applied at 4 °C overnight. After extensive washing in PBS/0.1% Triton X-100 (6 × 15 min), the samples were incubated in the secondary antibodies (*Supplementary file 2*) for 1 hr at room temperature. Unbound antibodies were removed in four changes of PBS /0.1%

Triton X-100 (15 min each) and nuclei were stained in 1:10,000 dilution of 10.9 mM DAPI (Invitrogen) for 30 min. After the final round of washes (3 × 10 min), the samples were coverslipped in Slow-Fade antifade medium (Invitrogen). Stacks of optical sections were taken using the Zeiss 880 confocal laser scanning microscope. Z-projections were generated in the Fiji image processing software (RRID:SCR_002285) and Imaris (Oxford Instruments, RRID:SCR_007370). Figures were constructed using Inkscape 1.1 (RRID:SCR_014479) and GIMP 2.10.30 (RRID:SCR_003182).

## Generation of BAC-reporters

The sox4-BAC was recombineered with different fluorescent reporters (*Supplementary file 3*) following the established protocol (*Buckley and Ettensohn, 2019*). The EL250 cells and the GFP recombination cassette were generous gifts from Dr. Buckley. The GFP coding sequence was replaced with either Kaede coding sequence or mCardinal coding sequence in the recombination cassette. Using the same protocol, we generated a sox2-Kaede BAC (*Supplementary file 3*). Embryos injected with recombineered BACs were observed at 2dpf, 4dpf, and 7dpf to confirm the expression pattern recapitulating sox2 or sox4.

## Microinjection of BACs

BAC-reporters were injected into fertilized eggs at a final concentration of 10 ng/μl as previously described (*Cheatle Jarvela and Hinman, 2014*). For double lineage tracing an equivalent amount of sox2-Kaede BAC and sox4-Cardinal BAC were mixed at a final concentration of 10 ng/μl. Injected positive embryos were sorted under fluorescent stereo microscope at 24–48 hpf and then were kept in sea water at 15 °C until future manipulations.

## Photo-conversion of Kaede *P. miniata* cells

Kaede is a photoconvertible fluorescent protein that changes irreversibly from green to red upon exposure to violet light (*Ando et al., 2002*). Photo-conversion of the Kaede was performed using an Andor Revolution XD spinning Disk confocal microscope emitting at 405 nm with 100% laser power for 60–80 s with snapshots taken every 10 s. After conversion transgenic embryos and larva were inspected for completion of conversion using the same microscope as quickly as possible to reduce the stress and transferred back to the plate with ASW.

## Live embryos/larva manipulation and live imaging

To trace cell fate we kept transgenic embryos and larvae individually in a small petri dish with ASW at 15 °C. Before imaging the animals were immobilized in 500 mM high-salt sea water for 1–2 min to remove the cilia, then mounted on the slides. Immediately after imaging, coverslip was carefully picked up and animals were transferred back to the dish. To increase survival rate the animals were manipulated as quickly as possible, the different laser power settings were used to minimize tissue damage in embryonic and larval stages. Regenerated larvae at different time points were imaged with consistent settings for comparison. Imaging was performed by using the Andor Revolution XD spinning Disk confocal microscope with Andor IQ3 system. Z-projections of stacks of optical sections were generated in the Fiji image software (RRID:SCR_002285). Figures were constructed using Inkscape 1.1(RRID:SCR_014479) and GIMP 2.10.30 (RRID:SCR_003182).

## Establishing Kaede protein stability in embryos

### Generation and injection of kaede mRNA

Kaede PCR product was amplified from the plasmid (RRID:Addgene_54726) for generating the template for capped mRNA synthesis (primers F: 5'- TAA TAC GAC TCA CTA TAG GGG TCG CCA CCA TGA GTC TGA T –3'; R: 5'-TTG CCG ATT TCG GCC TAT TGG –3') with mMESSAGE mMACHINE T7 transcription kit (ThermoFischer). Kaede mRNA was injected into the fertilized eggs at a final concentration of 300 ng/μl. Kaede-positive embryos were sorted under 488 nm fluorescent light at 15 hpf, then were kept in a plate with artificial sea water (ASW) at 15 °C until live imaging. These embryos were starved prior to imaging to avoid fluorescent background from their algal stomach contents.

### Generation of the conversion plot and the conversion rate plot

Photo-conversion was completed manually with Andor Revolution XD spinning Disk confocal microscope as described above. The time-lapse data were analyzed with Fiji. The pixel values (fluorescent

intensity, or FI) of 488 nm and 560 nm channels within selected areas were measured at each time point. The background FI was subtracted to obtain the net FI which was plotted over time to generate the photo-conversion plot. The ratio of net FI (560 nm) over net FI (488 nm) was calculated and plotted over time to generate the conversion rate plot (*Figure 3—source data 1*).

### Kaede stability - the fluorescence intensity curve

Twelve embryos injected with Kaede mRNA were photoconverted at 24 hpf and were imaged everyday up to 12dpf. Out of the 12 converted embryos, 6 survived the repetitive imaging session. The net FI of the 488 nm and 560 nm channels were collected and calculated following the steps described above. The net FI of both channels were plotted over time to generate the fluorescence intensity curve (*Figure 3—source data 1*).

## Quantification of green Sox4+ cells and statistical analyses

Quantification of green Sox4+ cells was conducted manually in larvae. Unhealthy, abnormal or dying larvae were not quantified. Quantification data of 5dpf larvae (Kaede converted at 2dpf embryonic stage, or embryonic conversion) and 7dpf larvae (Kaede converted at 4dpf larval stage, or larval conversion) were then processed for statistical analyses using the website https://www.estimation-stats.com/#/. Shared control comparison was performed using Hedge's g comparison (*Ho et al., 2019*). The confidence interval width was 95%. Hedge's g was chosen because it compares effect sizes across experiments and is corrected for small-sample bias. The p-value was calculated with the permutation T test.

The green Sox4+ cells at the regeneration leading edge were quantified manually in seven larvae at 3dpb using Fiji. The data was compared to the 2dpf embryonic conversion data. The effect size was Cliff's delta and the mean of data. The confidence interval width was 95%. Two side p-value was calculated with the Brunner-Munzel test. The data was compared to the 7dpf larval conversion data. Delta's cliff did not apply to these two groups. Instead, we compared the means of the two groups. The confidence interval width was 95%. Two side p-value was calculated with the Mann Whitney test (*Figure 4—source data 1*).

## Quantification of Kaede+ cells and determination of colors

Quantification of green, red and yellow Sox4+ cells was conducted manually in Z-stack images. The cellular area of each colored cell was compared to an adjacent background area with no Kaede signal to measure FI of 488 nm and 560 nm channels, respectively. The FI in the background area was subtracted from the FI at each channel within the cellular area to generate the net FI of the colored cell at 488 nm and 560 nm channels. We then calculated the ratio of net FI (560 nm) over net FI (488 nm) for each cell. The three colored groups were defined as follows: if the net FI ratio of 560 nm/488 nm >5, the cell is considered as a red cell; if the ratio is between Y and 5, the cell is considered as a yellow cell; if the ratio of 560 nm/488 nm< Y, the cell is considered as a green cell.

## Inhibition of cell division

To inhibit cell division, we used aphidicolin (Sigma A0781) at a dosage of 8.3 µg/g (25 µM) in sea water. To prevent drug elimination, we changed Aphidicolin and vehicle (DMSO) every 24 hr, so the larvae were continuously exposed. The first treatment was done immediately post-injury, followed by two more drug changes every 24 hr. Control animals received 0.08% DMSO (vehicle) incubation. The aphidicolin treatment, therefore, covered the phases of regeneration, and preceded the first observed sox2 and sox4 expression by at least 1 day. These treatment parameters were established after a series of pilot experiments using two drug concentrations (25 uM, 50 uM), followed by 1 hr incubation in 15 uM EdU, which was detected by standard protocol (invitrogen C10637). To elucidate sox2+, sox4+ cells fate without cell division the fertilized eggs were injected with sox2, sox4 Bacs, 1-week-old larvae were bisected, photoconverted, incubated in aphidicolin solution for 3 days and then imaged.

## Sample size of experiments

At least 30–50 embryos and/or larvae in three to five separate trials were analyzed in each in situ and immunostaining experiment. For quantification of FPs expression in live embryos and/or larvae were analyzed 7–10 samples from three to five separate matings for each injection of each BAC construct.

## Acknowledgements

The authors are grateful to Dr. Buckley for the assistance in BACs recombineering and Dr. Cheryl Telmer, Dr. Andrew Wolff, and Jonathan Andrade for helpful comments on the manuscript. The work was supported by NSF. IOS 1557431, NIH 1R24OD023046 and DSF Charitable Foundation.

## Additional information

### Funding

| Funder | Grant reference number | Author |
|---|---|---|
| National Science Foundation | NSF. IOS 1557431 | Veronica Hinman |
| National Institute of General Medical Sciences | NIH 1R24OD023046 | Veronica Hinman |
| DSF Charitable Foundation | | Veronica Hinman |

The funders had no role in study design, data collection and interpretation, or the decision to submit the work for publication.

### Author contributions

Minyan Zheng, Conceptualization, Data curation, Formal analysis, Methodology, Validation, Visualization, Writing – original draft, Writing – review and editing; Olga Zueva, Conceptualization, Data curation, Formal analysis, Investigation, Methodology, Validation, Visualization, Writing – original draft, Writing – review and editing; Veronica F Hinman, Conceptualization, Formal analysis, Funding acquisition, Investigation, Methodology, Project administration, Resources, Supervision, Writing – original draft, Writing – review and editing

### Author ORCIDs

Minyan Zheng (iD) http://orcid.org/0000-0001-7990-1773
Veronica F Hinman (iD) http://orcid.org/0000-0003-3414-1357

### Decision letter and Author response

Decision letter https://doi.org/10.7554/eLife.72983.sa1
Author response https://doi.org/10.7554/eLife.72983.sa2

## Additional files

### Supplementary files
- Supplementary file 1. Primers for probe amplification.
- Supplementary file 2. List of antibodies.
- Supplementary file 3. List of BAC.
- Transparent reporting form

### Data availability

All data generated in this study are included in the manuscript. Genomic sequence data is provided with accessing numbers and/or links to echinobase.org.

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
