## [Editor Report]

This manuscript presents a careful study of nervous system regeneration in the larval sea star using new transgenic tools for marking and following cells involved in regeneration. The authors find that these animals can regenerate their nervous system by the re-specification of existing cells, which are induced to express the embryonic neurogenesis program. The experimental approach is robust and creative and the data interpretation sound. For its contribution to our understanding of how cells are induced to contribute to specific cell lineages during regeneration, this work will be of interest to the broad community of researchers in regenerative and developmental biology.

---

## [Decision Letter]

**Decision letter after peer review:**

Thank you for submitting your article "Regeneration of the larval sea star nervous system by wounding induced respecification to the *Sox2* lineage" for consideration by *eLife*. Your article has been reviewed by 2 peer reviewers, and the evaluation has been overseen by Phillip Newmark as Reviewing Editor and Marianne Bronner as the Senior Editor. The reviewers have opted to remain anonymous.

Essential revisions:

1) In this carefully performed and thoughtfully presented manuscript, the only conclusion that requires either additional data or softening of claims is that new *Sox2* expression is induced in the absence of cell division. The results look suggestive of that, but Figure 6 only shows the effect on EdU labeling at 3 dpb (after 3 treatments with aphidicolin). Because lack of EdU labeling is shown only at 3 dpb, it is not clear to what extent proliferation is inhibited during the preceding 3 aphidicolin pulses. If there were any leakiness in the effects of the drug, some cells may still be dividing during the time at which *sox2*^+^ cells are specified. The authors' case would be strengthened if they show earlier time points to confirm lack of EdU incorporation during the initial stages of aphidicolin treatment. In the absence of such additional validation, the authors should soften their claims.

2) For all figures please include sample sizes (the numbers of larvae analyzed and the numbers of cells analyzed in each larva). For figures 5 and 6, it is unclear how many samples were analyzed.

3) The bottom panel of Figure 4F is not described in the figure legend or text. Please describe or delete it.

4) Reviewer #1's public review raised concerns about the description of the bac transgenesis system used here as "a lineage tracing system" and suggests replacing it with wording that more accurately reflects the true nature of these reporters. Please either revise the manuscript accordingly or provide a rationale for describing the system in this way.

5) Reviewer #1's public review asked for clarification about the infrequency of observing newly specified *sox2*^+^ cells that also express sox4 (Figure 5), so it would be helpful for the authors to provide additional interpretations about rarely observed colocalization patterns.

6) Reviewer #1's public review also suggested that the final paragraph of the discussion be revised to reflect the current, more nuanced and complex understanding of how stem cells, specialized/committed stem cells, and de- and trans-differentiating cells contribute to different animal models of regeneration, including Hydra and axolotl.

7) The authors should provide a supplementary table providing details on the length, sequence region, and accession number for each probe used for WMISH and FISH riboprobes.

8) The "Historic Kaede" label in red text in Figure 6 is difficult to read superimposed upon the red staining in the image. Please change the text to provide better contrast.

---

## [Author Response]

Essential revisions:1) In this carefully performed and thoughtfully presented manuscript, the only conclusion that requires either additional data or softening of claims is that new Sox2 expression is induced in the absence of cell division. The results look suggestive of that, but Figure 6 only shows the effect on EdU labeling at 3 dpb (after 3 treatments with aphidicolin). Because lack of EdU labeling is shown only at 3 dpb, it is not clear to what extent proliferation is inhibited during the preceding 3 aphidicolin pulses. If there were any leakiness in the effects of the drug, some cells may still be dividing during the time at which sox2^+^ cells are specified. The authors' case would be strengthened if they show earlier time points to confirm lack of EdU incorporation during the initial stages of aphidicolin treatment. In the absence of such additional validation, the authors should soften their claims.

We thank this reviewer for the positive comments. We agree that this is an important control, and have repeated this experiment to include earlier time points. Regenerating larvae were incubated continuously in aphidicolin for 24hrs with changes into new dishes with aphidicolin at each 24h. We also extended the time period out to 6 days. Thus larvae were incubated continuously for 1, 2, 3, 4, 5, or 6 days in aphidicolin that was never more than 24hr in diluted SW. These larvae were then treated with EdU to test for cell proliferation at each of these time points in addition to the 72h reported in the original manuscript. This new data is now presented as Supplementary Figure 6. This shows clearly that aphidicolin completely stops any cell proliferation as assayed by EdU staining. As a new finding, It also shows that the aphidicolin treated larvae do not regenerate the anterior structures as we see limited formation of anterior structures in aphidicolin treated compared to vehicle controls.

We added the following text into the methods section:

“We incubated regenerated larvae in the drug or vehicle for up to one week, with daily changes into fresh drug or vehicle, and analyzed them every 24h to test that Aphidicolin consistently blocked cell division over this time (Figure 6—figure supplement).”

2) For all figures please include sample sizes (the numbers of larvae analyzed and the numbers of cells analyzed in each larva). For figures 5 and 6, it is unclear how many samples were analyzed.

We have now included all of this information for each figure either directly in the figure plate or within the figure legend.

We also added the following text into the methods section:

“Sample size of experiments

At least 30-50 embryos and/or larvae in 3-5 separate trials were analyzed in each in situ and immunostaining experiment. For quantification of FPs expression in live embryos and/or larvae were analyzed 7-10 samples from 3-5 separate matings for each injection of each BAC construct.”

3) The bottom panel of Figure 4F is not described in the figure legend or text. Please describe or delete it.

We apologize for this oversight on our part. We have now added the following text in the figure legend to explain this part of the figure:

“(F) Quantification of Sox4 specification in embryos, larvae and regeneration. The Hedges' g for two comparisons against the embryonic group are shown in the above Cumming estimation plot. The raw data is plotted on the upper axes. On the lower axes, mean differences are plotted as bootstrap sampling distributions. Each mean difference is depicted as a dot. Each 95% confidence interval is indicated by the ends of the vertical error bars”

We have also added an additional reference to cite the method:

Moving beyond P values: data analysis with estimation graphics.

Joses Ho, Tayfun Tumkaya, Sameer Aryal, Hyungwon Choi, Adam Claridge-Chang

Nature Methods 16, 565–566 (2019). 10.1038/s41592-019-0470-3

And made modifications to the following paragraph into the methods section of the main text, which now reads:

“Quantification of Green Sox4+ cells and statistical analyses

Quantification of green Sox4+ cells was conducted manually in larvae. Unhealthy, abnormal or dying larvae were not quantified. Quantification data of 5dpf larvae (Kaede converted at 2dpf embryonic stage, or embryonic conversion) and 7dpf larvae (Kaede converted at 4dpf larval stage, or larval conversion) were then processed for statistical analyses using the website https://www.estimationstats.com/#/. Shared control comparison was performed using Hedge’s g comparison. The confidence interval width was 95%. Hedge’s g was chosen because it compares effect sizes across experiments and is corrected for small-sample bias. The p-value was calculated with the permutation T test.”

4) Reviewer #1's public review raised concerns about the description of the bac transgenesis system used here as "a lineage tracing system" and suggests replacing it with wording that more accurately reflects the true nature of these reporters. Please either revise the manuscript accordingly or provide a rationale for describing the system in this way.

The BAC reporter system that we use in this study works in the following way. BAC recombinant DNA is injected into the oocyte and after one to several rounds of embryonic cleavage, the DNA likely becomes integrated into the genome. The DNA is stably inherited by all of the progeny of this cell. Since the recombined fluorescent reporter is under the control of the endogenous basal promoter and native enhancer elements, the fluorescent reporter will express in the same spatio-temporal domain as the endogenous gene, but only in the clones of cells in which the DNA is inherited. We showed (Figure 3—figure supplement) that once a functional fluorescent protein is produced, it is very stable and will remain detectable above background for at least 7 days.

We therefore think that once FP is detected, it stably marks all of the lineage of the cell that expressed the gene – even after the gene is no longer being expressed. In this way it is a faithful marker of this lineage. We do agree with the reviewer’s comment that the FP will not trace the entire lineage of cells that express this gene at any time point as the original DNA reporter is mosaically inherited. This limitation can be overcome by examining enough larvae to ensure that all possible lineages are identified, but acknowledge that in our experiments, some *sox2*^+^ and soxc4+ lineages could certainly be missing from the analysis. We have therefore used this reviewers suggestion and made the following changes:

Abstract:

“In this study we develop new transgenic tools to follow the fate of populations of cells in the regenerating bipinnaria larva of the sea star Patira minaita.”

Introduction*:*

“We establish a novel photoconvertible BAC reporter system to trace populations of cells to determine the cellular origin of these regenerated neurons.”

Results*:*

“To test this hypothesis, we needed to establish a reporter system reporter that would distinguish new sox4+ cells from existing sox4+ cells and also follow their fate for several days. The BAC DNA construct is stably inherited by clones of cells and therefore present mosaically in the cells of the larvae. “

In multiple places we have also changed the term “lineages” to “cells” to better convey that the entire lineage is not examined.

Discussion, deleted the following sentence:

“To our knowledge, this is the first time that any cell lineage tracing studies have been performed in echinoderm regeneration.”

Added the following:

“The injected BAC DNA construct is stably inherited in clonal lineages of cells after one to several embryonic cell division and hence the reporter will express mosaically.”

5) Reviewer #1's public review asked for clarification about the infrequency of observing newly specified sox2^+^ cells that also express sox4 (Figure 5), so it would be helpful for the authors to provide additional interpretations about rarely observed colocalization patterns.

There are several reasons that we think that these cells are seen rarely. The first is likely because the BAC reporter system only labels lineages mosaically (as explained for point 4 above). Therefore in any one larvae, only some of the possible cells will be labelled. The second is that we think these cells are very rare. We designed the timing of the experiment to capture the possible newly expressing cells in the early process of specification, before excessive cell proliferation and think therefore that we are capturing the first positive cells and that these are rare. Many of the newly expressing *sox2*^+^ cells will form non neural ectoderm, and are therefore not expected to coexpress sox4.

We have made the following edits in the results to clarify this finding:

“We observe only few newly expressing sox2^+^ and sox4+ cells in this assay, likely as our reporter system is mosaic, and because there are in reality only small numbers of newly expressing sox2^+^ cells at this stage.”

6) Reviewer #1's public review also suggested that the final paragraph of the discussion be revised to reflect the current, more nuanced and complex understanding of how stem cells, specialized/committed stem cells, and de- and trans-differentiating cells contribute to different animal models of regeneration, including Hydra and axolotl.

We agree with the reviewer that we have not properly explained the complexity of mechanisms used in these model systems and have changed to final paragraph as below and have also included the relevant citations

“As new model systems are added to the study of regeneration, they present an opportunity to re-examine our understanding of regeneration and the ways in which cellular reprogramming is induced by wounding. […] We posit that echinoderms may have a natural capacity to maintain a cellular plasticity that permits induction of new lineage stem-cells, as well as an ability to use existing lineage-restricted stem cells and that this flexibility may underlie their extensive abilities for regeneration.”

7) The authors should provide a supplementary table providing details on the length, sequence region, and accession number for each probe used for WMISH and FISH riboprobes.

This information is now provided in Supplementary file1.

8) The "Historic Kaede" label in red text in Figure 6 is difficult to read superimposed upon the red staining in the image. Please change the text to provide better contrast.

The color on this text is now changed and should be more readable.